# CLUSTERED FEDERATED LEARNING WITH SLIGHTLY SKEWED LABELS

**Jiaming Pei**
School of Computer Science
The University of Sydney
jpei0906@uni.sydney.edu.au

**Wei Li**
School of Computer Science
The University of Sydney
weiwilson.li@sydney.edu.au

## ABSTRACT

The core of some clustered federated learning methods is the K-means algorithm, but it cannot identify which sample from which cluster under slightly skewed labels setting. This paper discussed this problem from the perspective of clustered federated learning and proposed a Gaussian mixture clustered method (GM-CFL) to measure the probability of the samples belonging to which clusters. This method efficiently aggregates the model parameters between clusters and is robust to non-IID data distribution. The empirical results demonstrate that our method outperforms other state-of-the-art clustered federated learning methods.

## 1 INTRODUCTION

Non-IID data heterogeneity in federated learning can manifest in various forms, one of which is skewed labels. Skewed label data can adversely impact the accuracy of federated learning models. To tackle this issue, clustered federated learning methods aim to group similar clients with similar model parameters. However, many K-means-based solutions are not sufficiently robust in handling slightly skewed labels in federated learning systems. Skewed labels can be measured by the number of missing labels among clients, where a missing label is a label that is not present in the client's data set. In this work, we define slightly skewed labels as cases where the number of missing labels per client does not exceed 10%. Figure 1 provides two examples illustrating the difference between skewed and slightly skewed labels.

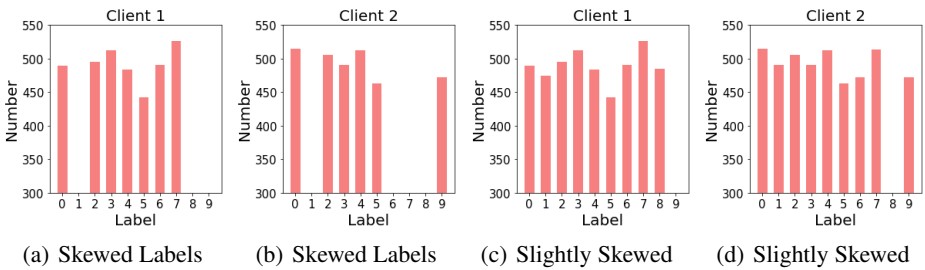

(a) Skewed Labels    (b) Skewed Labels    (c) Slightly Skewed    (d) Slightly Skewed

Figure 1: Skewed labels vs Slightly Skewed labels.

Not surprisingly, local model parameters could be similar, especially when datasets from different entities within the same business share common features McMahan et al. (2017); Wang et al. (2020); Liang et al. (2020); Zeng et al. (2022); Deng et al. (2020). In such cases, the K-means-based clustering method may require some assistance in splitting similar parameters to improve clustering performance. When dealing with coincident samples, it can also be challenging to differentiate them using K-means alone. In contrast, Gaussian mixture clustering is a compelling alternative that can return the probability of clustering assignment and provide an oval-shaped cluster, leading to better clustering performance, as illustrated in Figure 3. In this study, we explore the potential of Gaussian mixture clustering to divide coincident local model parameters arising from slightly skewed labels into distinct clusters. Before obtaining the ultimate global model, our proposed GMCFL method aggregates the local models to facilitate knowledge sharing within each cluster.

## 2 APPROACH

Figure 2 provides an overview of the proposed Gaussian Mixture Clustered Federated Learning (GMCFL) method. Due to slightly skewed labels, the parameters of local models are highly similar, leading to the appearance of numerous coincident points in Euclidean space. This situation hinders the clustering of model parameters into their respective clusters. Inspired by the Gaussian mixture equation $P(x) = \sum_1^K \alpha_i \cdot N(x|\mu_i, \Sigma_i)$, we assign weight $\alpha_i$ to each local model based on the probability of a sample belonging to a particular cluster. Next, we compute the expectation $\mu_i$ and covariance $\Sigma_i$ to determine the range of each cluster based on probability rather than the distance between samples and the cluster center. After clustering, the model parameters within each cluster are aggregated, and the aggregated model parameters are further aggregated among clusters. This process is repeated for multiple rounds until the global model converges.

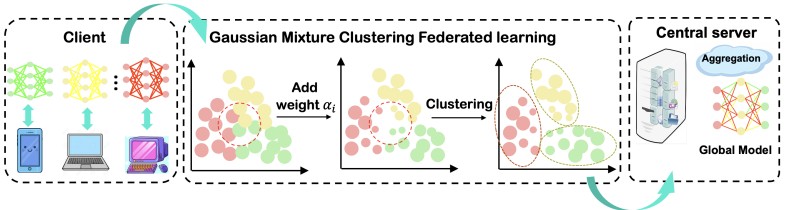

Figure 2: The workflow of the proposed GMCFL Method.

## 3 EXPERIMENTS

The MNIST and CIFAR-100 data set were used to test our method in the slight non-IID settings through randomly removing $\alpha\%$ label classes from training data set of each client. Table 1 shows that GMCFL is robust to different non-IID degrees and has acceptable accuracy on the image classification task. Figure 4 compares the performance of GMCFL with several state-of-the-art cluster Fl methods Liang et al. (2020); Pei et al. (2022b;a); Briggs et al. (2020). Since these algorithms are not well-equipped to handle coincident points between clusters, their accuracy suffers while dealing with slightly skewed labels.

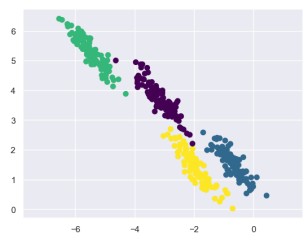

Figure 3: Gaussain mixture clusters when the number of clusters is four.

Table 1: The test accuracy of our method under different degree of skewed labels ($\alpha$)

| data sets | Acc($/\%$) |
| --- | --- |
| MNIST(10%) | 89.21 |
| CIFAR-100(2%) | 85.24 |
| CIFAR-100(5%) | 80.15 |
| CIFAR-100(10%) | 78.89 |

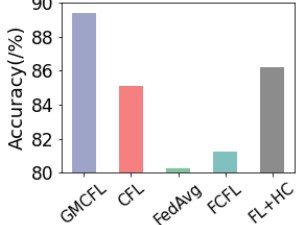

Figure 4: Comparison results when $\alpha$ is 10% on MNIST data set.

## 4 CONCLUSION AND FUTURE WORK

Our work aims to enhance the robustness of federated learning systems under slightly skewed labels. We propose the GMCFL method, which predicts the probability of coincident samples to achieve soft clustering. In future research, we will delve deeper into the reasons behind the suboptimal performance of the K-means-based CFL method in this context and further refine GMCFL to address other challenging non-IID scenarios.

URM STATEMENT

The authors acknowledge that at least one key author of this work meets the URM criteria of ICLR 2023 Tiny Papers Track.

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
