# OpenReview forum: "Clustered Federated Learning with Slightly Skewed Labels"
_ICLR.cc/2023/TinyPapers — Submitted to Tiny Papers @ ICLR 2023_

### Official Review · Reviewer_Y7fS · 2023-03-30

**Confidence:** 5

**Summary Of Contributions:**

The work discusses the setting of slightly skewed labels in clustered federated learning and proposes a Gaussian mixture clustering method to improve clustering performance. Experiments shows that this method efficiently aggregates the model parameters between clusters and is robust to slightly skewed labels.

**Rating:**

Needs Clarification (NC): a submission which does not meet the reviewing criteria and needs clarification for its described problem or solution

**Strengths And Weaknesses:**

**Strengths**

1. The proposed method is easy to understand, and readers may be able to reproduce it on their own.
2. The submission of this paper meets the formatting requirements and page limits.

**Weaknesses**

1. This work claims that existing clustered federated learning methods based on k-means are not robust enough to handle slightly skewed label scenarios, but they are usually aimed at more difficult non-IID scenarios. The explanation of existing methods in this paper is not convincing.
2. In the paper, $\alpha$ represents the average proportion of missing labels for each client, and the experiment was conducted on 10 clients using the MNIST dataset. It seems possible that 8 of the 10 clients have all categories, and only 2 clients each lack 5 categories? The results shown in Table 1 seem to reflect this, does this situation meet the slightly skewed labels proposed by the author? Moreover, why do various methods have such huge differences in the experimental scenario shown in Figure 4, which is very close to IID?

**Suggested Changes:**

1. Is there a bigger challenge in the scenario of slightly skewed labels that current clustered federated learning or federated learning techniques can not solve? This needs a more convincing explanation.
2. There seems to be a flaw in the definition of slightly skewed labels presented in this paper. The scenario where 8 out of 10 clients have all categories and only 2 clients each lack 5 categories still satisfies the definition given by the author.

---

### Official Review · Reviewer_vMV3 · 2023-04-02

**Confidence:** 4

**Summary Of Contributions:**

This paper proposes a new method called gaussian mixture clustered method (GM-CFL) to tickle the problem of clustering based federated learning under slightly skewed data. The proposed method can easily separate models in different clusters for aggregation purpose.

**Rating:**

Clear, Correct, and Reproducible (CCR): a submission which meets the reviewing criteria

**Strengths And Weaknesses:**

**Strengths:**

S1: This paper is original. The idea of using gaussian mixture clustering for federated learning has never been considered before.

S2: This paper is well written. The logic is easy to follow. All the findings in the paper is expressed clearly and effectively.

S3: The claims and conclusions are justified by the findings. Authors follow the formatting requirements.

**Weakness:**

W1: the quality of this paper is medium. Authors should show more experiments about why this method is useful and compare with the baseline clustering method.

W2: The concept about "Slightly Skewed" proposed the authors is minor. Why not consider a broader problem: clustered FL for skewed data?


**Suggested Changes:**

C1: For the setup of experiment, authors should state more clearly about the how the Non-IID data is partitioned and how the labels of each client is removed.

C2: In the approach section, authors should formalize their approach in a more clear and elegant way. Otherwise, it could be hard for readers to understand.

C3: Please consider to cite more related FL papers:

[1] Towards Federated Clustering: A Federated Fuzzy c-Means Algorithm (FFCM)

[2] Federated Multiple Label Hashing (FedMLH): Communication Efficient Federated Learning on Extreme Classification Tasks.

[3] An Efficient Framework for Clustered Federated Learning

---

### Meta-Review · Area_Chair_jKYe · 2023-04-04

**Recommendation:** Invite to archive
**Confidence:** 4

**Metareview:**

The paper considers the case of federated learning for clustering problems where the clients have slightly skewed labels.

**Summary:**

Although there are some concerns by the reviewers the paper is a good submission to this venue. It addresses the problem of handling skewed labels in federated learning settings.

**Reason For Not Giving A Higher Recommendation:**

Both the reviewers expressed concerns about how the parameter $\alpha$ is used. How the data/labels are partitioned and what clients can and cannot have certain labels to meet the "slightly skewed" criterion. Hope the authors can address these concerns.

**Reason For Not Giving A Lower Recommendation:**

The paper presents the problem it addresses clearly and experimental results are presented well.

---

### Decision · Program_Chairs · 2023-04-10

Invite to archive